# Phytochemical Compounds from *Laelia furfuracea* and Their Antioxidant and Anti-Inflammatory Activities

**DOI:** 10.3390/plants14040588

**Published:** 2025-02-14

**Authors:** Abimael López-Pérez, Luicita Lagunez-Rivera, Rodolfo Solano, Aracely Evangelina Chávez-Piña, Gabriela Soledad Barragán-Zarate, Manuel Jiménez-Estrada

**Affiliations:** 1Laboratorio de Extracción y Análisis de Productos Naturales Vegetales, Centro Interdisciplinario de Investigación para el Desarrollo Integral Regional Unidad Oaxaca, Instituto Politécnico Nacional, Hornos 1003, Santa Cruz Xoxocotlán 71230, Oaxaca, Mexico; ablopezp@ipn.mx (A.L.-P.); asolanog@ipn.mx (R.S.); gbarraganz@ipn.mx (G.S.B.-Z.); 2Laboratorio de Farmacología, Escuela Nacional de Medicina y Homeopatía, Instituto Politécnico Nacional, Guillermo Massieu Helguera, 239, La Escalera, Gustavo A. Madero 07320, Ciudad de México, Mexico; 3Instituto de Química, Universidad Nacional Autónoma de México, Circuito Exterior, Insurgentes Sur, C.U., Coyoacan 04510, Ciudad de México, Mexico; manuelj@unam.mx

**Keywords:** inflammation, *in vivo* evaluation, flavonoids, traditional medicine, orchids

## Abstract

*Laelia furfuracea* is an orchid endemic to Oaxaca, Mexico, used for the treatment of cough and has anticoagulant activity. We aimed to evaluate the anti-inflammatory and antioxidant activity of the hydroethanolic extract of *L. furfuracea* leaves and identify its phytochemical compounds. The leaf material was subjected to solid–liquid extraction. Compounds were identified by UPLC-ESI-qTOF-MS/MS. The Folin–Ciocalteu and aluminum trichloride methods were used to quantify phenols and flavonoids, respectively. The DPPH method was used to determine the antioxidant activity. The anti-inflammatory activity was evaluated in a model of carrageenan-induced plantar edema induced in Wistar rats. Compounds tentatively identified in *L. furfuracea* leaves were malic, citric, succinic, hydroximethylglutaric, azelaic, eucomic, and protocatechuic acids, saponarin, luteolin-7,3′-di-O-glucoside, isoorientin, and vitexin. The contents of total phenols and flavonoids and antioxidant activity were 394.7 ± 0.1 mg EqAG/g, 129.9 ± 0.005 mg EqQ/g, and 84.6 ± 1.4%, respectively. The anti-inflammatory effect of the extract was dose-dependent, where 1000 µg/paw presented a 43.4% reduction in inflammation, similar to naproxen. The anti-inflammatory and antioxidant effect of the hydroethanolic extract of *Laelia furfuracea* leaves was demonstrated. This effect may be due to the synergy between its compounds. This orchid is a potential candidate for future pharmacological research due to its anti-inflammatory activity.

## 1. Introduction

Inflammation is a hallmark of many diseases, including cancer, neurodegenerative diseases like Alzheimer’s, type II diabetes, rheumatoid arthritis, and asthma [1]. Anti-inflammatory therapy includes two groups, i.e., steroidal anti-inflammatories and nonsteroidal anti-inflammatory drugs (NSAIDs) [2]. NSAIDs have been a cornerstone in the management of various inflammatory, pain, and fever-related conditions. However, they have presented adverse effects, including those associated with cardiovascular, renal, and hepatic complications [1], as is the case with naproxen, which has presented side effects such as gastrointestinal and renal toxicity [3]. Therefore, currently a therapeutic alternative is the search for new compounds of natural origin in a safe, effective, and less harmful way to control inflammatory diseases.

Plants are an important source in the search for these new bioactive compounds, such as terpenes and phenolic compounds, mainly. They provide cellular protection against oxidative stress [4] that can otherwise lead to cell deterioration and death [5], causing acute, chronic, and degenerative inflammatory disease [6]. The discovery of specific molecules directed toward inflammatory agents, in a safe, effective, and less harmful manner, is the therapeutic strategy for the control of inflammatory diseases. Terpenes and flavonoids have beneficial effects on health due to their antioxidant [7], analgesic [8], and anti-inflammatory [9] properties.

These types of properties have been described recently in some Mexican orchids [10,11], such as the antinociceptive activity of *Cyrtopodium macrobulbon* (Lex.) G.A. Romero and Carnevali [12], vasorelaxant effect of *Trichocentrum brachyphyllum* (Lindl.) R. Jiménez [13], the anti-inflammatory and antioxidant activity of *Prosthechea michuacana* (Lex.) W.E. Higgins [14], and *P. karwinskii* (Mart.) J.M.H. Shaw [15,16]. Some species of the genus *Laelia* Lindl. are used in traditional medicine mainly by peasant communities [17]; for example, *Laelia autumnalis* (Lex.) Lindl. [18,19], *Laelia anceps* Lindl. [20,21], and *Laelia speciosa* (Kunth) Schltr. [21] are used to treat bumps and wounds. These orchids have also been studied to evaluate their biological properties. *L. anceps* presents cytotoxic activity in tumor cells of the central nervous system (U251), lung (SKLU-1) and breast (MCF-7) [22], in addition to having antihypertensive activity [20,21]. In *L. autumnalis,* its antihypertensive and vasorelaxant activity has been evaluated [18,19], while *L. speciosa* also presents vasorelaxant activity [21].

*Laelia furfuracea* Lindl. is an orchid endemic to Oaxaca, Mexico. In the Mixteca Alta of Oaxaca, it is known that this species is used as an herbal remedy for cough, preparing an infusion with its flowers, which is drunk as drinking water [22]. Groups of phytochemical compounds of pharmacological interest have been identified in this orchid, such as flavonoids and phenolic acids [23], and *in vitro* anticoagulant activity has been demonstrated by prolonging prothrombin (PT), activated partial thromboplastin (aPTT), and thrombin (TT) times [22,24,25]. Relating this effect to its chemical composition and its traditional use to alleviate an inflammatory process of the respiratory tract that is physiologically related to inflammation, we aimed to show evidence for the first time of the antioxidant and anti-inflammatory potential of this orchid and to identify its phytochemical composition, which alone or in synergy may be responsible for its biological effects. This is an opportunity for the search of new bioactive molecules with antioxidant and anti-inflammatory activity.

## 2. Results

### 2.1. Phytochemical Profile

Different groups of compounds were identified in the extract of *L. furfuracea*, such as tannins, flavonoids, saponins, cardiotonics, and quinones. Regarding fractions, tannins were the only group of compounds present in all fractions; flavonoids were not identified in the FC; saponins were present in the fractions of lower polarity (FH, FC, and FEA); quinones were present in the fractions of FC and FB; sesquiterpenlactones were identified only on FB; coumarins and alkaloids were not identified in the HELF and fractions (Table 1).

### 2.2. Compounds Identified by UPLC-ESI-qTOF-MS/MS

Chromatographic analysis tentatively identifies the presence of metabolites such as carboxylic acids, particularly citric, malic, succinic, hydroxymethylglutaric, and azelaic acids; a terpene identified was the iridoid glycoside; as to phenolic compounds of the phenolic acid groups, eucomic and protocatechuic acid were present. Some flavonoids identified were saponarin, luteolin-7,3′-di-O-glycoside, isoorientin, and vitexin, and flavonoid glucoside derived from isoorientin and vitexin (Figure 1, Table 2).

### 2.3. Total Flavonoids

The HELF presented a flavonoid content of 129.9 ± 0.01 mg EqQ/g with respect to the fractions. Those with the highest concentrations were the FC and FEA, with 198.1 ± 0.1 and 503.5 ± 0.2 mg EqQ/g, respectively (Table 3).

### 2.4. Total Phenols

In the same way, the FC and FEA fractions showed the highest values of total phenols, whose concentrations were 282.8 ± 0.1 and 612.5 ± 0.3 mg EqAG/g, while the HELF presented 394.7 ± 0.2 mg EqAG/g (Table 3).

### 2.5. Antioxidant Activity

Regarding the *in vitro* antioxidant activity by the DPPH technique, HELF presented 84.6 ± 1.4% inhibition of free radicals, with an AAI of 1.3 ± 0.02 mg EqAA/g. The FC and FEA fractions showed the best antioxidant activity. These values show *L. furfuracea* as an orchid with high antioxidant capacity (Table 3).

### 2.6. Anti-Inflammatory Activity

#### 2.6.1. Oral Administration

In the anti-inflammatory evaluation, HELF, administered orally to rats at different doses, did not present a significant dose–response effect (Figure 2a).

#### 2.6.2. Local Administration

On the other hand, the local administration of HELF, after six hours of evaluation, presented a significant dose–response anti-inflammatory effect compared to the control group. This anti-inflammatory effect of 43.7% was similar to the inhibition of inflammation caused by the administration of naproxen used as the reference drug, which had an effect of 45.7% (Figure 2b).

### 2.7. In Vivo Antioxidant Activity by Measurement of NO and GSH

The groups of rats treated with HELF with doses of 562.23 and 1000 µg/paw presented NaNO_2_ concentrations of 34.1 ± 1.8 µM and 28.2 ± 4.8 µM, respectively. Meanwhile, the concentration of NaNO_2_ in the animals in the control group was 28.0 ± 3.9 µM, with no significant differences between the values recorded for these groups.

In the measurement of GSH, the groups of animals that received the HELF doses did not present significant differences compared to the value presented by the control group.

## 3. Discussion

Chromatographic analysis identifies the presence of citric, malic, succinic, hydroxymethylglutaric, azelaic, eucomic, and protocatechuic acids, iridoid glycoside, saponarin, luteolin-7,3′-di-O-glycoside, isoorientin, and vitexin, and flavonoids glucoside derived from isoorientin and vitexin.

Out of the compounds identified, malic acid, eucomic acid, saponarin, and vitexin have been studied for their biological activities related to inflammation. Malic acid dependently decreases the content of tumor necrosis factor alpha (TNF-α), interferon γ (IFN-γ), and interleukins 6 (IL6) and IL10 [36]; besides, it reduces symptoms in rheumatic diseases [37]. Eucomic acid stimulates cytochrome *c* oxidase activity in the immortalized human keratinocyte cell line (HaCaT) [38]. Saponarin reduces the inflammatory response by inhibiting mitogen-activated protein kinase (MAPK) signaling, nuclear factor kappa β (NF-κB) activity, cytokine production, and expressions of marker genes specific for M1 polarization [39]; also, it presents hepatoprotective effects, increases the cellular antioxidant defense system, and levels of reduced glutathione (GSH) [40]. In the same way, it reduces the expression of inflammatory mediators such as IL-4 cytokines, IL-5, IL-13, phosphorylation of extracellular signal-regulated kinase (ERK), and p38 involved in the mitogen-activated protein kinase signaling pathway in RAW264.7 cells [41]. Vitexin inhibits cytokines IL-8, IL-17, IL-33, nitric oxide (NO), and monocyte chemoattractant protein-1 (MCP-1) and increases IL-10 [42]. We consider that the presence of these compounds in *L. furfuracea* is intervening to give the anti-inflammatory biological effect and antioxidant activity.

Regarding the antioxidant capacity of *L. furfuracea*, similar values were reported in the orchids *Dendrobium thyrsiflorum* Rchb. f. ex André [43], *D. tosaense* Makino, *D. moniliforme* (L.) Sw. [44], and *Eulophia ochreata* Lindl. [45]. This antioxidant percentage was higher in *L. furfuracea* than those reported in the extract by soxhlet with 56.6% [46] and by maceration 16.6% [47] of *Prosthechea karwinskii* and *Anacamptis pyramidalis* (L.) Rchb. f. with 54.1% [48]. With respect to the fractions, the percentage of inhibition of free radicals is higher; the FEA and FC of *L. furfuracea* presented 91.6 ± 0.2 and 82.9 ± 1.7%, and AAI of 2.1 ± 0.001 and 1.4 ± 0.006 mgEqAA/g, respectively. These fractions even turned out to be better antioxidants than the compound 4-hydroxy-3-methoxybenzyl alcohol isolated from *Gastrodia elata* Blume, which was 70% [49].

In this sense, it can be observed that there is a relationship in terms of antioxidant activity, total flavonoids, and phenols, both in the extract and in the fractions. This could be due to their composition, which has an impact on the fact that the antioxidant activity is a function of the number and position of the -OH groups, as is the case of phenols such as flavonoids, which have the highest number of hydroxyl groups and the highest rates of free radical inhibition [50].

Acute inflammation is a response of the body to repair a damaged area, which is characterized by redness, edema, heat, pain, and functional impotence [1]. This process triggers intracellular signals produced by immune cells, macrophages, and neutrophils responsible for the main inflammatory events. These are activated and bind to specific receptors to regulate cellular functions such as the expression of adhesion molecules, phagocytosis, cell death, and secretion of proinflammatory enzymes and chemotoxins, which are signs of an acute inflammatory response [1,2,3]. Therefore, considering that the anti-inflammatory effect in this study may be given by the phytochemical compounds of the leaf extract of *L. furfuracea*, their bioactivity depends not only on their molecular structure and concentration but also on other physiological factors in experimental animals [51]. Furthermore, structurally, phenolic compounds behave like acids, affecting solubility, glycosylation, and methylation [52], as is the case with some flavonoids linked to sugars that cause variations in their susceptibility to being digested, fermented, and absorbed in the gastrointestinal tract. However, only a part of these compounds can cross the intestinal wall to be located in the areas of the injury to act as anti-inflammatory [53].

Regarding the safety of consuming *L. furfuracea*, its use is only known in traditional medicine, where peasant communities prepare an infusion using three flowers of the plant to treat cough, but an *in vivo* study has not been carried out to demonstrate its biosafety.

On the other hand, in local administration, *L. furfuracea* showed an anti-inflammatory effect. Comparing this anti-inflammatory effect with other orchids such as *Eulophia ochreata* [54] and *Prosthechea karwinskii* [15], they also present a similar dose–response anti-inflammatory effect. Still, the leaf extract of *L. furfuracea* inhibits inflammation as a protective response, perhaps enhanced by the antioxidant action of its compounds. These compounds are likely to act on the inflammation system, since they also act on the coagulation model [23], and this model converges with inflammation [55], thus supporting its anti-inflammatory effect.

With respect to the fractions evaluated, none of these (FH, FC, FAE, FB, and FA) presented inhibition of edema in the animals, so it is inferred that the synergy of one or more compounds present plays an important role in achieving or potentiating their bioactivity [56] when the crude extract was evaluated.

In the same way, in the group of animals treated with HELF, it was observed that the concentrations of NO and GSH did not increase, so these endogenous antioxidant systems were not directly responsible for the inhibition of edema. This may be due to other enzymatic mechanisms that could work together to counteract oxidative stress, such as superoxide dismutase (SOD), catalase (CAT), and glutathione peroxidase (GPx) [57]. However, it could be that the phenolic compounds in *L. furfuracea* extract play an important role as exogenous antioxidants that complement crucial endogenous systems, maintaining a balance that contributes to antioxidant and anti-inflammatory protection, as well as the prevention of related conditions with oxidative stress [58].

## 4. Materials and Methods

### 4.1. Vegetal Material

The leaf material of *L. furfuracea* was collected in the municipality Santo Domingo Yanhuitlán, Oaxaca, México, in an oak forest with *Juniperus* and *Arbutus*. For this purpose, permission was obtained from the authorities of the Commissariat of Communal Property, as well as a scientific collection permit granted by the Mexican Ministry of Environment and Natural Resources to RS (02228/17). A voucher specimen (R. Solano 4244) was herborized and deposited in the OAX Herbarium of the Centro Interdisciplinario de Investigación para el Desarrollo Integral Regional (CIIDIR) of Oaxaca.

### 4.2. General Experimental Procedures

#### 4.2.1. Reagents

Ascorbic acid (CAS 50-81-7), 1,1-Diphenyl-2-picryl-hydrazyl (CAS 1898664), AlCl_3_ (CAS 7446-70-0), Folin–Denis Reagent (CAS 47742), gallic acid (CAS 149-91-7), quercetin (CAS 7446-70-0), naproxen (CAS 26159-34-2), and carrageenan (CAS 9000-07-1) were purchased from Sigma Aldrich (Toluca, Mexico).

#### 4.2.2. Extraction Process

The leaf material was dehydrated at 40 °C until constant weight, then pulverized in a mill (IKA^®^ M20, Northchase Parkway, Wilmington, DE, USA) and passed through a physical test sieve with an opening of 250 microns, mesh number 60 (MOTINOX, Toluca de Lerdo, Ciudad de México, Mexico), to have a homogeneous particle size and facilitate its extraction. The material was subjected to a solid–liquid extraction in a Soxhlet equipment (Puebla, Puebla, Mexico), using 5 g of leaf material with water–ethanol (1:1) as a solvent, at 78.2 °C for two hours. The extract was filtered and concentrated in a rotary evaporator (BÜCHI, R-210, Darmstadt, Germany). Finally, the hydroethanolic extract of *L. furfuracea* (HELF) was subjected to a liquid–liquid extraction by increasing polarities to obtain fractions of hexane (FH), chloroform (FC), ethyl acetate (FEA), butanol (FB), and water (FA) [23] (Appendix A).

#### 4.2.3. Characterization of the Phytochemical Profile

The phytochemical profile of the extract and fractions was obtained following the procedure to determine its main groups of metabolites [59] (Appendix A).

#### 4.2.4. Analysis of Compounds by UPLC-ESI-qTOF-MS/MS

One mg of *L. furfuracea* leaves extract was dissolved in 1 mL of water with 10% methanol. Subsequently, it was filtered through a 0.45 µm nylon filter. Ten µL of the filtered solution was taken and mixed with 90 µL of acetonitrile. The injection volume for the analysis was 10 µL.

The analysis of compounds was performed using an ultra-high-performance liquid chromatography system (UPLC, Thermo Scientific, Ultimate 3000) coupled to an Impact II mass spectrometer (Bruker Daltonics, Billerica, MA, USA), equipped with electrospray ionization and quadrupole with time-of-flight (UPLC-ESI-qTOF-MS/MS). This analysis was based on a previous study [10]. The column used was a Thermo Scientific Acclaim 120 C18 (2.2 µm, 120 Å, 50 × 2.1 mm). The mobile phase was A: 0.1% formic acid in water and B: acetonitrile. The gradient system was set up as follows: 0% B (0–2 min), 1% B (2–3 min), 3% B (3–4 min), 32% B (4–5 min), 36% B (5–6 min), 40% B (6–8 min), 45% B (8–9 min), 80% B (9–11 min), and 0% B (12–14 min) at a flow rate of 0.35 mL/min. The mass spectrometer was operated in negative electrospray mode at 0.4 Bar (5.8 psi), with a mass range of 50–1000 *m*/*z*, in MS/MS mode, and an ionization capillary voltage of 2700 V. Data were processed using the Data Analysis software 3.1 for mass spectrometry (Bruker Daltonics). The compounds were tentatively identified by comparing their mass spectra with Bruker MetaboBase, Plant metabolites, MassBank libraries, and those reported in previous articles.

#### 4.2.5. Quantification of Total Flavonoids

The determination of total flavonoids was carried out with the aluminum trichloride (AlCl_3_) colorimetric method [60], using quercetin as a standard for the calibration curve. The results were expressed as mg quercetin equivalents/g sample (mg EqQ/g).

#### 4.2.6. Quantification of Total Phenols

The concentration of total phenols was determined by the method of Folin and Ciocalteau [61], using gallic acid as a standard for the calibration curve. The results were expressed as mg equivalent of gallic acid/g of sample (mg EqAG/g).

#### 4.2.7. *In Vitro* Antioxidant Activity

The antioxidant activity was determined with the free radical scavenging method by 1,1-diphenyl-2-picryl-hydrazyl (DPPH), using ascorbic acid as a standard for the calibration curve, calculating the antioxidant activity index (IAA) with the concentration of DPPH between IC50, as well as the percentage of inhibition of the DPPH radical, as recommended by Ibarra [62].

#### 4.2.8. Evaluation of Anti-Inflammatory Activity

##### Experimental Animals

Female Wistar rats, 7–9 weeks old and weighing 180 ± 20 g, were used from the vivarium of the Center for Research and Advanced Studies (CINVESTAV, Mexico City). The animals were housed under regular conditions at a temperature of from 22 to 24 °C in a 12-h light–dark cycle. Their diet consisted of standard laboratory food with free access to food, and purified water *ad libitum*, kept in 43 × 53 × 20 glass acrylic boxes with removable iron mesh lid. The designated area was 187 cm^2^ per animal, and between 4 and 6 individuals were placed in each cage. Efforts were made to minimize animal suffering and to reduce the number of animals used. Each rat was used in only one experiment and euthanized in a CO_2_ chamber at the end of the assay [63]. The conditions of the rats were according to the Official Mexican Standard for the Care and Management of Experimental Animals (NOM-062-ZOO-1999) and the regulations of the International Council for Laboratory Animal Science (ICLAS) [64].

##### Design of Experiments

The animals were randomly divided into groups for oral and local administration of HELF. In oral administration, groups of rats with 11 individuals were used: the positive control received 300 mg/kg p.o. naproxen (reference drug); the negative control received distilled water (vehicle); and the treatments received 562.23 mg/kg p.o. and 1000 mg/kg p.o. of the HELF. Treatment was administered with a gastric cannula one hour before inducing inflammation. Before oral administration, the animals were fasted for 8 h and had free access to water.

In local administration, ten groups of rats with 6 subjects were used: 1000 μg/paw of naproxen (positive control group), distilled water (vehicle control group), and three groups that received HELF in doses of 300, 562.23, and 1000 μg/paw; in five other groups, each of the extract fractions (FH, FC, FEA, FB, and FA) was administered in doses of 1000 μg/paw. The doses evaluated were selected based on an increased logarithmic scale. Each fraction was dissolved in saline with 10% Tween 80 as a vehicle. The administration of the volume was parenteral and plantar, one hour before inducing inflammation in the rats. The animals did not require prior treatment; two hours after administering the study agents, the animals received water and, three hours later, food. Inflammation was induced using the plantar edema model, applying an injection of 100 μL of 1% carrageenan in saline to the right hind paw of each rat. The volume of the paw where the inflammatory process was induced was measured with a digital plethysmometer (LE 7500 Mca. Panlab Harvard/Apparatus, Hill Road Holliston, USA) at 0, 30, 60, 120, 180, 240, 300, and 360 min after administering carrageenan. The percentage of inhibition of inflammation was calculated from the differences in the basal volume between the different times, using the following equation:% inhibition = (ABC_control_ − ABC_treated_/ABC_control_)100(1)

The area under the curve (AUC) was calculated through the following equation:ABC = ∑ ((Δ*vol* B_greater_ + Δ*vol* b_smaller_) h/2)(2)Δ*vol* Inflamed—*vol* basal(3)

##### *In Vivo* Antioxidant Activity (NO and GSH)


*Nitric oxide (NO) in the paw*


The concentration of NO was determined in the homogenates of the paws of the rats from the negative control and basal groups and those that presented anti-inflammatory activity when the extract was administered at doses of 562.23 and 1000 μg/paw locally. The Cayman colorimetric test for nitrates/nitrites and the Griess technique were used to determine NO [65]. The concentration of sodium nitrite (NaNO_2_) in the paw was expressed as µmol/g of tissue.


*Reduced glutathione (GSH) in the paw*


GSH was quantified in the same way in the rats in the negative control and basal groups and in those where there was inhibition of inflammation with HELF at doses of 562.23 and 1000 μg/paw locally, using the Ellman reagent method (5,5’-dithio-bis-(2-nitrobenzoic acid) known as DTNB (Cayman, 69-78-3, Ellsworth Rd. Ann Arbor, MI, USA), which allows detecting thiols. The readings were carried out at 412 nm. The GSH concentration in the paw was expressed as nM/g of tissue [65]. All determinations were performed in triplicate.

### 4.3. Statistical Analysis

The results of the anti-inflammatory evaluation were expressed as the mean and standard error for each group of animals. An analysis of variance (ANOVA) implemented in GraphPad Prism 5 was applied, as well as a Newman–Keuls multiple comparison test of means to determine the differences between them. Values were considered statistically significant when *p* < 0.05.

## 5. Conclusions

*Laelia furfuracea* is an orchid with a dose-dependent antioxidant and anti-inflammatory effect. It is possible that its activity depends on the synergy of its compounds or any of these, such as carboxylic acids, terpenes, and phenolic compounds, mainly flavonoids and phenolic acids present in its leaf structures.

A relationship was observed between the antioxidant activity *in vitro* with the concentration of phenols and flavonoids, both in the crude extract and in the ethyl acetate fraction of the leaves.

The antioxidant and anti-inflammatory effects, chemical composition, and traditional use for alleviating inflammatory process of the respiratory tract–physiologically related to inflammation– support our findings on the antioxidant and anti-inflammatory potential of *L. furfuracea* for the first time. This represents an alternative for future research studies in the search for new bioactive compounds with pharmacological activity

## Figures and Tables

**Figure 1 plants-14-00588-f001:**
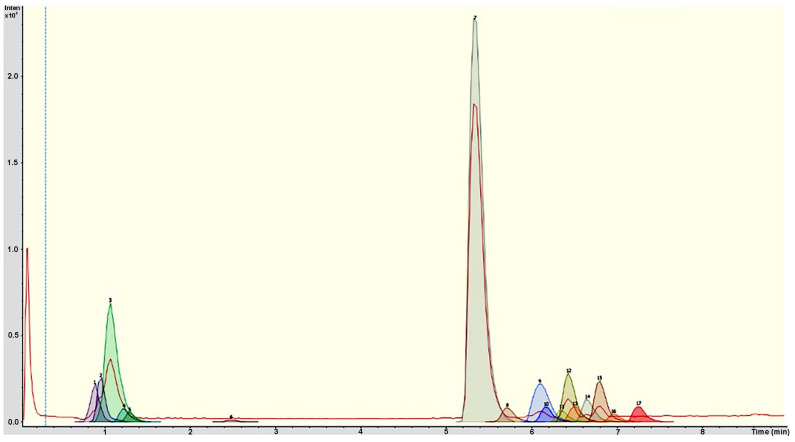
UPLC-ESI-qTOF-MS/MS chromatogram of the *Laelia furfuracea* leaf extract. The red line represents the baseline chromatogram. The identified compounds are listed as follows: 1 (unidentified), 2 (malic acid), 3 (citric acid), 4 (succinic acid), 5 (hydroxymethylglutaric acid), 6 (procatechuic acid), 7 (eucomic acid), 8 (luteolin-7,3′-di-O-glucoside), 9 (saponarin), 10 (isoorientin), 11 (vitexin), 12 (unidentified), 13 (flavonoid glucoside derived from isoorientin), 14 (unidentified), 15 (flavonoid glucoside derived from vitexin), 16 (iridoid glucoside), and 17 (azelaic acid).

**Figure 2 plants-14-00588-f002:**
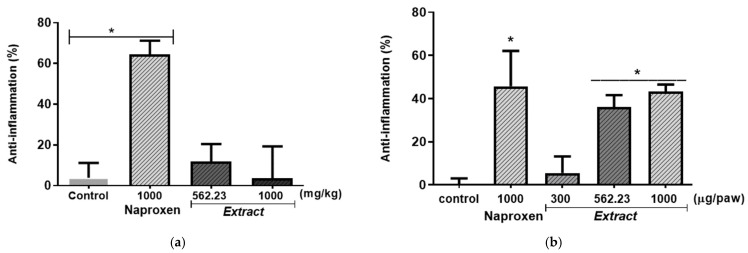
Anti-inflammatory effect at different doses of the hydroethanolic extract of *Laelia furfuracea* leaves in the model of plantar edema with carrageenan. (**a**) oral administration, the values were reported in mg/kg; (**b**) local administration, the values were reported in µg/paw. Each column represents the mean ± SD (n = 6); * indicates a significant difference between the treatments *p* < 0.05 of the control group (saline solution), compared to the positive control group (naproxen) and the extract at different doses.

**Table 1 plants-14-00588-t001:** Phytochemical profile of the extract and fractions of *Laelia furfuracea* leaves.

Phytochemical Tests	Sample					
	HELF	FH	FC	FEA	FB	FA
Flavonoids	+	+	−	+	+	+
Saponins	+	+	+	+	−	−
Tannins	+	+	+	+	+	+
Cardiotonics	+	−	−	+	+	+
Coumarins	−	−	−	−	−	−
Sesquiterpenlactones	−	−	−	−	+	−
Quinones	+	−	+	−	+	−
Alkaloids	−	−	−	−	−	−

Hydroethanolic extract of *Laelia furfuracea* (HELF); fractions: hexane (FH), chloroform (FC), ethyl acetate (FEA), butanol (FB), water (FA), presence (+), and absence (−).

**Table 2 plants-14-00588-t002:** Tentative identification of compounds in the hydroethanolic extract of *Laelia furfuracea* leaves by UPLC-ESI-qTOF-MS/MS.

PN	RT (min)	Error (ppm)	M/Z [M-H]^−^	Collision Energy (ev)	Fragments MS/MS *	Chemical Formula	Compound	Chemical Class	IR
1	0.9	5.6	165.0405	16.6	75.0082, 129.0180, 147.0265	C_9_H_10_O_3_	Unknown	Organic acid	[26]
2	0.9	5.0	133.0142	15.8	115.0040, 71.0137	C_4_H_6_O_5_	Malic acid	Hydroxydicarboxylic acid	[26,27,28,29]
3	1.1	0.1	191.0197	17.3	111.0090, 87.0089, 85.0296, 129.0194, 173.0085	C_6_H_8_O_7_	Citric Acid	Tricarboxylic acid	[26,29]
4	1.2	0.8	117.0192	15.4	73.0292, 99.0097	C_4_H_6_O_4_	Succinic acid	Dicarboxylic acid	[26]
5	1.3	0.7	161.0457	16.5	99.0448	C_6_H_10_O_5_	Hydroxymethylglutaric acid	Dicarboxylic acid	[26,29]
6	2.5	1.4	153.0191	16.3	109.0297	C_7_H_6_O_4_	Procatechuic acid	Phenol	[26,28,30]
7	5.3	2.4	239.0555	18.5	179.0346, 177.0554, 149.0604, 133.0660, 107.0496, 195.0662	C_11_H_12_O_6_	Eucomic acid	Phenolic monocarboxylic acid	[27,31]
8	5.7	1.2	609.1452	28.3	447.0918, 448.0962	C_27_H_30_O_16_	Luteolin-7,3′-di-O-glycoside	Glycoside flavone	[26]
9	6.0	1.9	593.1501	28.1	311.0551, 431.0980, 297.0406	C_27_H_30_O_15_	Saponarin	Flavonoid	[26,32]
10	6.1	3.1	447.0925	23.7	357.0618, 327.0504, 429.0813, 285.0405, 297.0416	C_21_H_20_O_11_	Isoorientin	Hydroxyflavone	[26]
11	6.3	1.5	431.0977	23.3	311.0557, 283.0612, 341.0634	C_21_H_20_O_10_	Vitexin	Hydroxyflavone	[26,29]
12	6.4	2.1	449.2019	23.7	269.1394, 270.1417, 225.1492, 209.1185, 251.1284, 89.0232, 287.1479	C_20_H_34_O_11_	Unknown	Flavonoid glycoside	[29,30,32]
13	6.5	1.8	591.1336	28.1	327.0500, 357.0610, 447.0921, 489.1032	C_27_H_28_O_15_	Flavonoid glycoside and isoorientin derivative	Flavonoid	[26,28,33]
14	6.5	2.6	413.1442	22.8	269.1018, 99.0455, 101.0244, 125.0250	C_19_H_26_O_10_	Unidentified	-	-
15	6.6	2.0	575.1395	27.3	311.0555, 341.0660, 431.0982, 473.1075, 513.1362, 263.0798, 161.0449	C_27_H_28_O_14_	Flavonoid glycoside and vitexin derivative	Flavonoid	[26,28]
16	6.8	n.c.	435.2225	23.4	389.2201	Unknown	Iridoid glycoside	Terpene	[34,35]
17	7.0	6.8	187.0989	17.2	125.0962, 141.8669, 169.0867	C_9_H_16_O_4_	Azelaic acid	Dicarboxylic acid	[26]

PN: peak number, RT: retention time, IR: identification reference, n.c.: not calculated. * The fragments were ordered according to their intensity, starting with those of greater height.

**Table 3 plants-14-00588-t003:** Antioxidant activity, flavonoids, and total phenolics of *Laelia furfuracea* leaf extract and fraction.

Tests	Total Flavonoids (mg EqQ/g)	Total Phenolics (mg EqAG/g)	Antioxidant Activity	
			Inhibition (%)	AAI (mg EqAA/g)
HELF	129.900 ± 0.010	394.700 ± 0.200	84.600 ± 1.400	1.300 ± 0.020
FH	81.900 ± 0.020	201.000 ± 0.200	50.100 ± 1.400	0.800 ± 0.010
FC	198.100 ± 0.100	282.800 ± 0.100	82.900 ± 1.700	1.400 ± 0.006
FEA	503.500 ± 0.200	612.500 ± 0.300	91.600 ± 0.200	2.100 ± 0.001
FB	118.100 ± 0.040	275.200 ± 0.100	52.100 ± 0.400	0.900 ± 0.027
FA	40.700 ± 0.020	207.800 ± 0.100	23.400 ± 1.100	0.300 ± 0.010

Hydroethanolic extract of *Laelia furfuracea* (HELF); fractions: hexane (FH), chloroform (FC), ethyl acetate (FEA), butanol (FB), water (FA), and AAI (antioxidant activity index); mg EqAA (milligram equivalents of ascorbic acid); mg eqQ (milligram equivalents of quercetin); and mg eqAG (milligram equivalents of gallic acid).

## Data Availability

Data are contained within the article.

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
