# Peer review of "Phytochemical Compounds from Laelia furfuracea and Their Antioxidant and Anti-Inflammatory Activities"

_plants, 2025, doi:10.3390/plants14040588_

Round 1
Reviewer 1 Report
Comments and Suggestions for Authors
General comment:
The study evaluated the anti-inflammatory and antioxidant activities of the hydroethanolic extract from the leaves of the orchid Laelia furfuracea and identified its phytochemical compounds. Analyses revealed the presence of organic acids (malic, citric, succinic) and flavonoids (saponarin, luteolin-7,3'-di-O-glucoside), with significant total phenolic and flavonoid contents and an antioxidant activity of 84.6%. The extract exhibited an anti-inflammatory effect comparable to naproxen, reducing inflammation by 43.4% at a dose of 1000 μg/paw. These results suggest that Laelia furfuracea has pharmacological potential due to the synergy among its active compounds.
Suggestions:
- The introduction is well-documented and provides a solid background for the study but could benefit from a more cohesive structure and a clearer emphasis on the knowledge gaps and the unique contributions of the research.
- The animal experiments are well-described, with clearly defined groups and specific doses. The method of inducing inflammation (carrageenan-induced paw edema) and the use of plethysmometry to measure volume are widely accepted standards for this type of study. However, I suggest:
a) Providing a rationale for the chosen doses of 300 mg/kg, 562.23 mg/kg, and 1000 mg/kg.
b) Explaining the acclimatization process for the animals.
c) Justifying the selection of the number of animals. Why did some groups have 6 animals, while others had 11? - Adding additional details regarding animal housing, such as the type of food administered, feeding frequency and quantity, the temperature conditions maintained, the number of animals per cage, their gender, age, and weight.
- Including a graphical representation of the experimental protocol to enhance clarity.
- Clarifying why the authors interpreted values as (mg/kg) in Figure 2(a) and (µg/paw) in Figure 2(b). An explanation of this in the figure legend would help readers less familiar with the subject understand it more easily.
- Adding a paragraph in the discussion section about the safety profile of Laelia furfuracea.
- A thorough review of the text for formatting issues is recommended. For example: (a) Line 97: L. furfuracea should be italicized. (b) Line 64: in vitro should be italicized, etc.
Author Response
Reply to the review report (Reviewer 1)
Comments 1: [The introduction is well-documented and provides a solid background for the study but could benefit from a more cohesive structure and a clearer emphasis on the knowledge gaps and the unique contributions of the research.]
Response 1: [Thank you for pointing this out. We agree with this comment. Therefore, we have changed the introduction according to your recommendation]”[This change is marked in red on paragraphs 1, 3, and 4, on lines 36-45, 55-57, and 73-77, respectively.]
Comments 2: [The animal experiments are well-described, with clearly defined groups and specific doses. The method of inducing inflammation (carrageenan-induced paw edema) and the use of plethysmometry to measure volume are widely accepted standards for this type of study. However, I suggest:
a) Providing a rationale for the chosen doses of 300 mg/kg, 562.23 mg/kg, and 1000 mg/kg.
b) Explaining the acclimatization process for the animals.
c) Justifying the selection of the number of animals. Why did some groups have 6 animals, while others had 11?.]
Response 2: [Thank you for pointing this out. We agree with this comment. Therefore, we have explained the following.]”
- a) The doses evaluated in this study were determined based on a logarithmic scale, the logarithmic scale shows the base value 10 raised to the power of a value. In this case, logarithm is represented as Log, it designates the power to which a number can be raised. Record. 1 is shown as 101= 10, in this case, 10 has a logarithm of 1 because 10 raised to the power of 1 is 10; in the case of Log. 2 is 102 = 100, where 10x10 =100. Record. 3 is 103 where 10x10x10 = 1000. The logarithm is used in different sciences. In chemistry, in pharmacology the logarithmic scale is used to calculate the concentration and time of action of chemical substances such as drugs, compounds of natural origin, or other types of substances in bioassays. Log stairs were used in this study 2.5, 2.75, and 3, to determine concentrations of 300, 562.23, and 1000 µg/kg in oral administration or mg/paw when administered locally. These concentrations change about time, in pharmacokinetics it is observed that from the moment the chemical substance is administered, and as it is absorbed, it appears in the blood until reaching a maximum concentration (Cmax) at a certain time (Tmax). This plasma concentration is important since its derivative of it determines the relationship between the dose (μg, mg, g) and the volume in which it is found (mL, L).
- b) Animals were acclimatized to our facilities during 2 weeks since they were obtained from the vivarium, they were acclimated to a controlled temperature of 22-24°C and 12-hour light/dark cycles. After two hours, and once acclimatized, the animals were provided with free access to food and purified water ad libitum.
- c) The number of animals in the oral and local administration was different, in the oral administration 11 individuals were used per group and in the local administration 6. This information was clarified in the methodology section on lines 309 and 315.]
Comments 3: [Adding additional details regarding animal housing, such as the type of food administered, feeding frequency and quantity, the temperature conditions maintained, the number of animals per cage, their gender, age, and weight.]
Response 3: [Thank you for pointing this out. We agree with this comment. Therefore, we have added additional details about regarding animal]”[This information is on line 295-303].
Comments 4: [Including a graphical representation of the experimental protocol to enhance clarity.]
Response 4: [Thank you for pointing this out. We agree with this comment. Therefore, we have included a graphical representation of the experimental protocol]”[ This information is outlined in Annex 2].
Comments 5: [Clarifying why the authors interpreted values as (mg/kg) in Figure 2(a) and (µg/paw) in Figure 2(b). An explanation of this in the figure legend would help readers less familiar with the subject understand it more easily.]
Response 5: [Thank you for pointing this out. We agree with this comment. As we evaluated two different ways of administering the extract (local and oral), the doses evaluated were different. The title of Figure 2 was modified for its correct understanding, also we have explained the values reported in oral 2 (a), and local 2 (b) administration.]”[This change is marked in red on lines 140-142].
Comments 6: [Adding a paragraph in the discussion section about the safety profile of Laelia furfuracea.]
Response 6: [Thank you for pointing this out. We agree with this comment. However, the only thing known about Laelia furfuracea that guides for its safe consumption is its traditional use. On the other hand, we evaluated the cytotoxicity of the extract (0.01 - 1.0 mg/mL) using the MTT test (we did not consider including the results of this test in the article) ensuring the safety of the extract for further research and potential use. But a study of its biosafety in animals has not been carried out, but it is important to consider it in another study and demonstrate its safe consumption.]”[We have added a paragraph discussing the safety of L. furfuracea in the line 206-208.]
Comments 7: [A thorough review of the text for formatting issues is recommended. For example: (a) Line 97: L. furfuracea should be italicized. (b) Line 64: in vitro should be italicized, etc.]
Response 7: [Thank you for pointing this out. We agree with this comment. Therefore, we have reviewed all manuscript to detect these formatting errors.]”[This change is marked in red on lines 71,82,119,121,146,287,334.]
Reviewer 2 Report
Comments and Suggestions for Authors
· The title needs to be more informative and attractive, for instance the authors only mention the anti-inflammatory activity although the study evaluates the antioxidant activity and identifies the phytochemical compounds.
· The abstract, please refer to the instructions for authors “We strongly encourage authors to use the following style of structured abstracts, but without headings”.
· Introduction, from line 39 to 42, please add the biological activities of these plants .
· Please, update the old references in the introduction.
· Line from 66 to 76 is more suitable for the dissection part.
· “2.5. Antioxidan activity”, please, correct the spelling.
· Table 2, please, follow the journal style and refer to the instructions for authors.
· Conclusion, please, add future perspectives of the field.
· Graphical abstract is highly recommended.
· English editing is highly recommended.
· Recommended reference to improve the manuscript´s introduction: Elrasoul, A.S.A., Mousa, A.A., Orabi, S.H., Mohamed, M.A.E.G., Gad-Allah, S.M., Almeer, R., Abdel-Daim, M.M., Khalifa, S.A., El-Seedi, H.R. and Eldaim, M.A.A., (2020). Antioxidant, Anti-Inflammatory, and Anti-Apoptotic Effects of Azolla pinnata Ethanolic Extract against Lead-Induced Hepatotoxicity in Rats. Antioxidants, 9(10), p.1014.
Comments on the Quality of English LanguageEnglish editing is highly recommended.
Author Response
Author's Reply to the Review Report (Reviewer 2)
Comments 1: [The title needs to be more informative and attractive, for instance the authors only mention the anti-inflammatory activity although the study evaluates the antioxidant activity ad identifies the phytochemical compounds.]
Response 1: [Thank you for pointing this out. We agree with this comment. Therefore, we have changed the title to something more informative and attractive. Which integrates anti-inflammatory and antioxidant activity, and the identification of phytochemical compounds. At the same time, the keyword antioxidant activity was eliminated so as not to repeat it with the title, and orchid was added. ][This change is marked in red on lines 2-3, and line 33.]
Comments 2: [The abstract, please refer to the instructions for authors “We strongly encourage authors to use the following style of structured abstracts, but without headings”.]
Response 2: [Thank you for pointing this out. We agree with this comment. Therefore, we have modified the abstract according to the structure of the journal, without the headings.]”[In addition, minor changes were made marked in red on line 18]
Comments 3: [Introduction, from line 39 to 42, please add the biological activities of these plants.]
Response 3: [Thank you for pointing this out. We agree with this comment. Therefore, We have added only the biological activities that relate to the objective of this study.]”[This change is marked in red on lines 55-57.]
Comments 4: [Please, update the old references in the introduction.]
Response 4: [Thank you for pointing this out. We agree with this comment. Therefore, we have updated the old references in the introduction.]”[This change is marked in red, and the list of references was ordered according to their appearance in the text.]
Comments 5: [Line from 66 to 76 is more suitable for the dissection or discussion part.]
Response 5: [Thank you for pointing this out. We agree with this comment. Therefore, we have changed this paragraph to the discussion part.]”[ This change is marked in red, and some of the information is on the line 191-197.]”
Comments 6: [“2.5. Antioxidan activity”, please, correct the spelling.]
Response 6: [Thank you for pointing this out. We agree with this comment. Therefore, we have changed this correctly.]”[ This change is marked in red, on line 118 ]”
Comments 7: [Table 2, please, follow the journal style and refer to the instructions for authors.]
Response 7: [Thank you for pointing this out. We agree with this comment. Therefore, we have changed this correctly, following the style of the magazine and the instructions for authors. We have eliminated the references described at the bottom of the table, marked in red as indicated in the following table, and can be seen in the table of the manuscript.]
17
7.0
6.8
187.0989
17.2
125.0962, 141.8669, 169.0867
C9H16O4
Azelaic acid
Dicarboxylic acid
[37]
PN: Peak number, RT: Retention time, IR: Identification reference. * The fragments were ordered according to their intensity, starting with those of greater height. 37: Massbank; 38: El-Hawary et al., 2020; 39: Ghareeb et al., 2018; 40: Heffels et al., 2017; 41: Llorach et al., 2019; 42: Ma et al., 2023; 43: Ragheb et al., 2023; 44: Sun et al., 2023; 45: Zhang Qun-Qun et al., 2016; 46: Zhao et al., 2022.
Comments 8: [Conclusion, please, add future perspectives of the field.]
Response 8: [Thank you for pointing this out. We agree with this comment. Therefore, we have added future perspectives of the field]” [This is marked in red, on lines 363-367.]
Comments 9: [Graphical abstract is highly recommended.]
Response 9: [Thank you for pointing this out. We agree with this comment. Therefore, we have sent the graphical abstract in a separate file]” [We have added the graphical abstract in annex 1.]
Comments 10: [English editing is highly recommended.]
Response 10: [Thank you for pointing this out. We agree with this comment. The English edition was reviewed by a specialist]
Comments 11: [Recommended reference to improve the manuscript´s introduction: Elrasoul, A.S.A., Mousa, A.A., Orabi, S.H., Mohamed, M.A.E.G., Gad-Allah, S.M., Almeer, R., Abdel-Daim, M.M., Khalifa, S.A., El-Seedi, H.R. and Eldaim, M.A.A., (2020). Antioxidant, Anti-Inflammatory, and Anti-Apoptotic Effects of Azolla pinnata Ethanolic Extract against Lead-Induced Hepatotoxicity in Rats. Antioxidants, 9(10), p.1014.]
Response 11: [Thank you for pointing this out. We agree with this comment. Therefore, we have taken into consideration your recommended reference to improve the introduction of the manuscript.] ”[ This change is marked in red on paragraphs 1, 3, and 4, on lines 36-45, 55-57, and 73-77, respectively.]”

Reviewer 3 Report
Comments and Suggestions for Authors
A brief summary
The paper describes the chemical analysis and the anti-inflammatory and antioxidant activity of the hydroethanolic extract of the Mexican orchid Laelia furfuracea. UPLC-ESI-MSMS-qTOF was used for identification of compounds, the Folin-Ciocalteu and aluminum trichloride methods were used to quantify phenols and flavonoids, and DPPH method was used to determine the antioxidant activity. The anti-inflammatory activity was evaluated in a model of plantar edema induced with carrageenan in Wistar rats.
General concept comments
The manuscript appears to be clear. It scientifically sounds and the experimental design is appropriate. Figures and tables, as well as statistical analysis, are appropriate. The conclusions are consistent with the results. The cited references are relevant, and the do not include an excessive number of self-citations. Anyway, In any case, a smaller number of self-citations would be preferable.
Specific comments
The manuscript appears to have been written a little hastily. The scientific names should be italicized throughout the manuscript. Inaccuracies and typographical errors must be corrected in all parts of the manuscript. Significant figures in table data should be checked. References should be checked according to the rules of the journal.
Figure 2.p<0.05 or p ≤ 0.05?
Materials and Methods:
Geographical references of the place of collection of plant material are missing
Sigma Aldrich : town and state are missing
The method used to describe the phytochemical profile of the extract and fractions (line 261) should be described, at least in outline form, obviously with the reference
AlCl3: Numbers must be in subscript format.
Comments on the Quality of English LanguageThe English could be improved to more clearly express the research.
Author Response
Author's Reply to the Review Report (Reviewer 3)
Comments 1: [The manuscript appears to be clear. It scientifically sounds and the experimental design is appropriate. Figures and tables, as well as statistical analysis, are appropriate. The conclusions are consistent with the results. The cited references are relevant, and the do not include an excessive number of self-citations. Anyway, In any case, a smaller number of self-citations would be preferable.]
Response 1: [Thank you for pointing this out. We agree with this comment. Therefore, we wanted to add a smaller number of self-citations, so as not to restore value to the information presented here, it was impossible. Most of the studies in this group of orchids that support this research have been developed by our working group. There is little information on medicinal uses and biological activities in Mexican orchids particularly in Laelia furfuracea. We have been pioneers in the biological study of this species, which due to its findings has potential in research to search for new bioactive compounds.]
Comments 2: [The manuscript appears to have been written a little hastily. The scientific names should be italicized throughout the manuscript. Inaccuracies and typographical errors must be corrected in all parts of the manuscript. Significant figures in table data should be checked. References should be checked according to the rules of the journal.]
Response 2: [Thank you for pointing this out. We agree with this comment. Therefore, we have reviewed all manuscript to detect these formatting errors, and we have changed this correctly, following the style of the magazine and the instructions for authors.]”[This change is marked in red on lines 71,82,119,121,146,287,334, and Table 3]
Comments 3: [Figure 2.p<0.05 or p ≤ 0.05?]
Response 3: [Thank you for pointing this out. We agree with this comment. Therefore, we have changed the term correctly] ”[ This change is marked in red on lines 143 and 354]”
Comments 4: [Materials and Methods:
Geographical references of the place of collection of plant material are missing
Sigma Aldrich : town and state are missing
The method used to describe the phytochemical profile of the extract and fractions (line 261) should be described, at least in outline form, obviously with the reference
AlCl3: Numbers must be in subscript format.]
Response 4: [Thank you for pointing this out. We agree with this comment. Therefore, we have added information.] ”[This is marked in red]
Geographical references of the place of collection of plant material are missing, lines 233-235
Sigma Aldrich: town and state are missing, line 246
The method used to describe the phytochemical profile of the extract and fractions (line 261) should be described, at least in outline form, obviously with the reference. This information is outlined in Annex 2
AlCl3 and NaNO2 : Numbers must be in the subscript format, lines 148, 149,243,281,340]